# IKKβ Inhibition Attenuates Epithelial Mesenchymal Transition of Human Stem Cell-Derived Retinal Pigment Epithelium

**DOI:** 10.3390/cells12081155

**Published:** 2023-04-13

**Authors:** Srinivasa R. Sripathi, Ming-Wen Hu, Ravi Chakra Turaga, Rebekah Mikeasky, Ganesh Satyanarayana, Jie Cheng, Yukan Duan, Julien Maruotti, Karl J. Wahlin, Cynthia A. Berlinicke, Jiang Qian, Noriko Esumi, Donald J. Zack

**Affiliations:** 1Department of Ophthalmology, Wilmer Eye Institute, The Johns Hopkins University School of Medicine, Baltimore, MD 21205, USA; 2Henderson Ocular Stem Cell Laboratory, Retina Foundation of the Southwest, Dallas, TX 75231, USA; 3Caris Life Sciences, 350 W Washington St., Tempe, AZ 85281, USA; 4Department of Biology, Georgia State University, Atlanta, GA 30303, USA; 5Emory Eye Center, Department of Ophthalmology, Emory University, Atlanta, GA 30322, USA; 6Research and Development, Phenocell, 06130 Grasse, France; 7Shiley Eye Institute, University of California, San Diego, CA 92093, USA; 8Solomon H. Snyder Department of Neuroscience, The Johns Hopkins University School of Medicine, Baltimore, MD 21205, USA; 9Department of Molecular Biology and Genetics, The Johns Hopkins University School of Medicine, Baltimore, MD 21205, USA; 10Department of Genetic Medicine, The Johns Hopkins University School of Medicine, Baltimore, MD 21205, USA; 11Institute for NanoBioTechnology, Johns Hopkins University, Whiting School of Engineering, Baltimore, MD 21218, USA

**Keywords:** stem cells, differentiation, retinal pigment epithelium, epithelial-mesenchymal transition, TGF–β/–α, kinase inhibitors, transcriptomics, PVR and AMD

## Abstract

Epithelial-mesenchymal transition (EMT), which is well known for its role in embryonic development, malignant transformation, and tumor progression, has also been implicated in a variety of retinal diseases, including proliferative vitreoretinopathy (PVR), age-related macular degeneration (AMD), and diabetic retinopathy. EMT of the retinal pigment epithelium (RPE), although important in the pathogenesis of these retinal conditions, is not well understood at the molecular level. We and others have shown that a variety of molecules, including the co-treatment of human stem cell-derived RPE monolayer cultures with transforming growth factor beta (TGF–β) and the inflammatory cytokine tumor necrosis factor alpha (TNF–α), can induce RPE–EMT; however, small molecule inhibitors of RPE–EMT have been less well studied. Here, we demonstrate that BAY651942, a small molecule inhibitor of nuclear factor kapa-B kinase subunit beta (IKKβ) that selectively targets NF-κB signaling, can modulate TGF–β/TNF–α-induced RPE–EMT. Next, we performed RNA-seq studies on BAY651942 treated hRPE monolayers to dissect altered biological pathways and signaling events. Further, we validated the effect of IKKβ inhibition on RPE–EMT-associated factors using a second IKKβ inhibitor, BMS345541, with RPE monolayers derived from an independent stem cell line. Our data highlights the fact that pharmacological inhibition of RPE–EMT restores RPE identity and may provide a promising approach for treating retinal diseases that involve RPE dedifferentiation and EMT.

## 1. Introduction

The retinal pigment epithelium (RPE), which consists of a highly pigmented, hexagonally packed cuboidal monolayer of epithelial cells, is essential for vision. Healthy RPE cells exhibit marked apicobasal polarity, with their apical surface interdigitating the outer segments of the light-sensing cells of the retina (rod and cone photoreceptors) and their basal surface resting on the basement membrane, known as Bruch’s membrane, which separates the RPE from its blood supply, the choriocapillaris. RPE cells promote the health and function of photoreceptor cells in multiple ways, including the transfer of oxygen and nutrients, phagocytosis of rod outer segments, synthesis and recycling of vitamin-A metabolites, and protection against reactive oxygen species [1,2]. RPE dysfunction and cell death, which occur in a number of retinal diseases, generally leads to photoreceptor degeneration and vision loss [3]. As one of the pathological processes causing RPE dysfunction, epithelial-to-mesenchymal transition (EMT) has increasingly drawn the attention of vision researchers [4,5,6,7,8]. During EMT progression, RPE cells lose their polarity, cellular adhesions and pigmentation, and acquire a fibroblast-like phenotype, resulting in a dysfunctional state that can no longer properly support the health and function of photoreceptor cells [9]. Additionally, the EMT-associated contractile activity of RPE cells can lead to complex retinal detachments that are difficult to treat. 

It has been shown that injury and stress can induce a variety of pathological changes in the RPE, including trans-differentiation and EMT [5,7,10,11,12], and that RPE–EMT is involved in multiple retinal diseases, including proliferative vitreoretinopathy (PVR) [4,6,13], neo-vascular (“wet”) age-related macular degeneration (AMD) [14], and atrophic (“dry”) AMD [8,15,16]. To better understand the pathogenesis and help develop therapeutic strategies for these diseases, we have been studying the process of RPE–EMT using human stem cell-derived RPE cells (hRPE) [10,11,12]. As an in vitro model of EMT, we and others have been using the co-treatment of hRPE monolayer cultures with TGF–β and TNF–α (TGF–β/TNF–α) to analyze the molecular changes that occur during the progression of EMT and to test the effects of molecules that attenuate EMT [7,10,11]. In our previous studies, using transcriptomic and proteomic datasets, we have used “upstream regulator analysis” (URA) to predict potential upstream targets of RPE–EMT, such as transcription factors, kinases, cytokines, growth factors, and microRNAs [10,11]. Among the predicted regulators were several components of the nuclear factor kappa B (NF-κB) signaling pathway.

NF-κB, a master regulator of inflammation and immune responses, is one of the signaling pathways that promote EMT by directly regulating several EMT transcription factors (EMT–TFs) and mesenchymal genes [17,18,19]. The NF-κB family is composed of five structural members, including NFKB1 (p50), NFKB2 (p52), RELA (p65), RELB, and REL [20]. In resting cells, NF-κB exists as a dimer in the cytoplasm in association with the κB (IκB) protein inhibitor. Diverse stimuli induce the activation of the IκB kinase (IKK) complex, which phosphorylates IκB proteins and consequently leads to their proteasome-mediated degradation. Released NF-κB dimers translocate to the nucleus, where they bind to their preferred DNA sequences in the promoters or enhancers of their target genes and activate transcription [21]. The IKK complex is a key regulatory node of the NF-κB cascade, which is composed of two catalytic subunits (IKKα and IKKβ) and a regulatory subunit (IKKγ), and is modulated by post-translational events including the phosphorylation, ubiquitination, and degradation of its inhibitory subunit [22,23,24]. NF-κB signaling is constitutively activated in a range of malignancies [25], and contributes to their metastatic potential through the induction and maintenance of EMT [18,19,26,27]. Dysregulated NF-κB signaling modulates sub–RPE drusen deposits in hiPS RPE cells containing AMD risk alleles [28].

Based on our above-mentioned prediction of RPE–EMT regulators, we tested several candidate small molecules that target these dysregulated pathways for their ability to modulate and reduce RPE–EMT induced by TGF–β/TNF–α. Here, we demonstrate that BAY651942, an ATP competitive potent inhibitor of IKKβ that selectively targets NF-κB signaling activity [29,30], can inhibit RPE–EMT. To validate this finding, we also tested the effect of BMS345541, another IKKβ inhibitor that blocks NF-κB-dependent transcription [31], and found that it also could modulate RPE–EMT. Additionally, we performed transcriptomic (RNA-seq) analysis to dissect the altered molecular events and associated biological pathways elicited by BAY651942. Taken together, our results show that IKKβ inhibition attenuates the transcriptomic and physiological changes that are associated with TGF–β/TNF–α-induced RPE-EMT, and thus identify the inhibition of IKKβ as a potential approach for treating retinal diseases that involve RPE–EMT.

## 2. Materials and Methods

### 2.1. Human Pluripotent/Embryonic Stem Cell (hiPSC/hESC) Cultures and RPE Differentiation

The human embryonic stem cell (hESC) line H7 and human induced pluripotent stem cell (hiPSC) line was obtained under license from WiCell (Madison, WI, USA). The hiPSC EP1 was generated as described previously [32]. Use of these cell lines has been authorized by the Johns Hopkins University (JHU) Institutional Stem Cell Research Oversight Committee (ISCRO 00000023). All policies and guidelines mandated by the ISCRO were strictly followed for the data generated in this manuscript. hiPSC/hESC lines were cultured and differentiated into RPE as previously described [33,34]. Briefly, the hiPSC (EP1, IMR90.4) and hES (H7) lines were maintained on Matrigel basement membrane matrix (Corning, Bedford, MA, USA) coated tissue culture plates with mTeSR1 medium (Stem Cell Technologies, Vancouver, Canada) in 5% O_2_ and 10% CO_2_ conditions and amplified by clonal propagation using the ROCK pathway inhibitor blebbistatin (Sigma-Aldrich, St. Louis, MO, USA). For hRPE differentiation, stem cells were plated at a higher density (25,000 cells per cm^2^) and maintained in mTeSR1 to form a monolayer. The culture medium was replaced with a differentiation medium (DM) for 7 weeks. Differentiating cells were enzymatically dissociated to make single-cell suspension using 0.25% (wt/vol) collagenase IV (Gibco) and resuspended in AccuMAX (Sigma-Aldrich, St. Louis, MO, USA). Cells were replated on to Matrigel basement membrane matrix coated plates and maintained in an RPE medium [70% DMEM, 30% Ham’s F12 nutrient mix, 2% B27-serum-free supplement, 1% antibiotic-antimycotic solution (Invitrogen, Waltham, MA, USA)] for ~3 months to form mature RPE monolayers.

### 2.2. A549 and H1299 Cell Cultures and EMT Induction In Vitro

Human non-small cell lung cancer cells A549 (CCL-185^TM^) and H1299 (CRL-5803™) were cultured in RPMI-1640 supplemented with 10% fetal bovine serum (FBS) in a 5% CO_2_ incubator at 37 °C. 90% confluent cells were used for all the experiments. Inducing TGF–β signaling-associated cancer-EMT in A549 and H1299 cells were treated with 20 ng/mL of recombinant human TGF–β1 in RPMI-1640 for 48 h in 37 °C/5% CO_2_ conditions.

### 2.3. Small-Molecule Treatment

The small molecule kinase inhibitors BAY651942 (Bayer, Wuppertal, Germany) and BMS345541 (Selleckchem, Houston, TX, USA) were reconstituted in DMSO (10 mM and 200 µM stocks), loaded into Echo qualified 384-well polypropylene microplates, and dispensed precisely in multiples of 2.5 nL droplets at the desired concentrations in triplicate using an ECHO 550 acoustic liquid handling system (Labcyte, San Jose, CA, USA).

### 2.4. RPE–EMT Induction In Vitro

For inducing TGF–β signaling-associated RPE–EMT in hRPE monolayers, cells were co-treated with 20 ng/mL of recombinant human TGF–β1 (Thermo Scientific, Waltham, MA, USA; Catalog # PHG9204) and recombinant human TNF–α protein (R&D systems, Minneapolis, MN, USA Catalog # 210-TA-020) in RPE medium for 24 h in 37 °C/5% CO_2_ conditions. For inducing EMT in A549 and H1299 lines, cells were treated with 20 ng/mL of recombinant human TGF–β1 only.

### 2.5. RNA Isolation and Quantitative RT-PCR (qRT-PCR) 

Total RNA from hRPE monolayers was extracted using an Isolate II RNA mini kit (Bioline, Taunton, MA, USA). cDNA was synthesized by reverse-transcription (High Capacity cDNA kit; Applied Biosystem). qRT-PCRs were performed using SsoAdvanced Universal SYBR Green Supermix (Bio-Rad, Hercules, CA, USA). All samples were run in biological triplicate, and expression levels were normalized by the geometric mean of three housekeeping genes (CREBBP, GAPDH and SRP72) (the geometric mean, which reduces the effect of outlier values, was calculated by multiplying together the Ct values of each of the housekeeping genes and then talking the cube root of the resulting product) [35]. Gene-specific primers sequences used for this study are included in Appendix A.

### 2.6. RNA-Seq and Data Processing

First-strand cDNA synthesis was performed with 200 ng total RNA using anchored oligo-dT and SuperScript III First-Strand Synthesis with SuperMix (ThermoFisher, Waltham, MA, USA). Second-strand cDNA synthesis was performed using RNase H, DNA Polymerase I, and Invitrogen Second Strand Buffer (ThermoFisher, Waltham, MA, USA). Double-stranded cDNA was purified using DNA Clean & Concentrator-5 (Zymo Research, Irvine, CA, USA). Library preparation was performed using the Nextera XT DNA Library Preparation Kit (Illumina, San Diego, CA, USA). Libraries were cleaned using Agencourt AMPure XP beads according to manufacturer’s instructions (Beckman Coulter, Brea, CA, USA). Libraries were evaluated by the High Sensitivity DNA Kit on a 2100 Bioanalyzer. They were then multiplexed and sequenced on an Illumina HiSeq with 50 bp paired-end reads. Reads were aligned to NCBI build 37.2 using TopHat (v2.1.0) [36]. Cuffquant and Cuffnorm (Cufflinks v2.2.1) were used to quantify expression levels and calculate normalized FPKM values [37].

### 2.7. RNA-Seq Data Analysis

We performed a Student’s *t*-test to identify differentially expressed genes (DEGs). For the TGF–β/TNF–α-induced EMT data set, DEGs were defined as genes with log2 fold change >1 and adjusted *p* < 0.1. For unsupervised hierarchical clustering, the Pearson correlation coefficient was used to construct a linkage matrix, and Ward’s method was used to calculate the distance between clusters.

### 2.8. Biological Pathway and Upstream Regulator Analysis

Pathway analysis was performed using Ingenuity Pathway Analysis (IPA) (QIAGEN, Redwood City, CA, USA). Transcripts identified to be differentially expressed during RPE–EMT compared to untreated monolayers with >2-fold change and *p* < 0.05 were input into IPA and KEGG (Kyoto Encyclopedia of Genes and Genomics) for bioinformatics analysis using gene IDs. Differentially-expressed transcripts were analyzed in IPA using core analysis followed by comparison analysis between dissociation time course, TGF–β/TNF–α concentration, and BAY-651942 treatment. Data sets were assessed for the prediction of canonical pathways and upstream regulators [38].

### 2.9. Hierarchical Transcription Factors Analysis

To identify the relationships between differentially expressed genes (DEGs) and their potential regulators, we integrated our gene expression data with available RPE chromatin accessibility data using a previously described method [39,40,41]. In the analysis, we only considered DEGs with open chromatin accessibility. To identify the enriched transcription factor (TF) binding motifs in the open regions, we utilized the “findMotifsGenome.pl” module in HOMER [42] with RPE ATAC-seq data as the background. We generated the required bed files, including the genomic coordinates for DEGs and the background, using a custom code. We executed the following command in HOMER: “findMotifsGenome.pl DEG.bed hg19 output -size 5000 -bg RPE_ATAC.bed.” To perform connectivity analysis of the DEGs, we used the STRING database [43] with default parameters.

### 2.10. Statistical Analysis

All statistical analyzes were performed using a Python statistical function (scipy.stats) and library (statsmodels). Fisher’s exact test, Student’s *t*-test, one-way ANOVA, and Pearson’s correlation coefficient were used to assess the significance. Fold change, *p*-values and false discovery rate (FDR) were calculated in the analysis.

## 3. Results

### 3.1. IKKβ Inhibitor (BAY651942) Treatment Partially Inhibits TGF–β/TNF–α-Induced RPE–EMT and Restores Expression of RPE Genes

Given the role of RPE–EMT in PVR and possibly in AMD, efforts have been made to identify inhibitors of RPE–EMT for potential therapeutic use [44,45]. Despite some progress in this field [46,47], since a safe and effective RPE–EMT inhibitor has yet to be reported, we sought to utilize our previously reported RPE–EMT model and transcriptional studies [10,11,12] to identify a suitable inhibitor. Our previous transcriptomic [11], proteomic [10], and phosphoproteomic [12] studies identified several malignancy-associated EMT factors that were also significantly altered during RPE–EMT. Although malignancy-associated EMT is controlled by a variety of biological pathways, NF-κB signaling is activated in a wide range of human cancers, and is involved in maintaining epithelial cell plasticity and metastasis via the orchestrated modulation of IKK-2/IκBα/NF-κB [26]. Multiple tumor-associated studies have shown that IKKβ is a key factor that regulates metastasis, and that the NF-κB signaling pathway is targeted by IKKβ inhibitors [48,49,50]. Moreover, the Upstream Regulator Analysis (URA) module, which generated kinase expression patterns that we delineated in our previous transcriptome study [11] suggested activated IKKβ expression during TGF–β/TNF–α-induced EMT (Figure 1D). Collectively, these findings led us to further investigate IKKβ inhibition as an approach to modulate RPE–EMT. 

First, to confirm the effect of the IKKβ inhibitor BAY651942 [51] on cancer-related EMT, we tested the drug on two well-established non-small cell lung cancer lines, A549 and H1299, which have been shown to have high basal levels of NF-κB activity [52,53]. We initially tried to induce EMT by co-treatment with TGF–β/TNF–α (each at 20 ng/mL), but found that this treatment induced significant cell death with both the A549 and H1299 cell lines; however, upon further testing we found treatment with TGF–β alone (20 ng/mL) induced EMT without affecting cell survival. We treated both cell lines with either TGF–β alone or a combination of TGF–β and BAY651942. Next, we analyzed the differential expression of multiple EMT markers by qPCR analysis. Our data show that *CDH1* (E-cadherin) levels markedly increased with the BAY651942 treatment in both lines (Figure 1A,B). We also found that the expression of EMT–related factors SNAIL (*SNAI1*) and SLUG (*SNAI2)* was suppressed in both lines (Figure 1A,B). The downstream EMT-associated factors vimentin (*VIM*) and alpha smooth muscle actin (α-SMA) showed a small, but not notably significant, decrease in expression with drug treatment (Figure 1A,B). Mesenchymal markers were markedly upregulated in TGF–β treated cells along with a reduction in the epithelial markers. This effect was reversed upon treatment with BAY651942, leading to a more epithelial-like cell state. Interestingly, we also observed markedly increased CDH1 expression in A549 cells upon treatment with BAY651942 compared to the control (i.e., not treated with TGF–β), which may reflect higher basal levels of NF-κB activity that could lead to latent or low-grade EMT at the cells’ basal state. Taken together, these studies are consistent with the notion that IKKβ inhibition modulates tumorigenesis-associated EMT, and led us to investigate the possibility that BAY651942 treatment could also inhibit RPE–EMT.

Using our TGF–β/TNF–α-induced RPE–EMT model, we treated hRPE monolayer cultures with BAY651942 and used qPCR to assess potential effects on the expression of EMT-related and RPE differentiation-related genes (Figure 1C). BAY651942 treatment restored the expression of key RPE factors, including *BEST1* (12-fold), *RPE65* (3-fold), *CRX* (5-fold) (Figure 1E), *L-RAT* (6-fold), *RLBP1* (4-fold), *TYR* (2-fold), and *MITF* (2-fold) (Figure 2E). Concomitantly, it also significantly suppressed the expression of multiple EMT factors, including *SNAI1* (1.6-fold), *JAG1* (3-fold), *HMGA2* (4-fold), *CDH2* (2-fold), and *FN1* (11-fold) (Figure 1F). Additionally, as expected since IKKβ inhibition inhibits NF-κB signaling, BAY651942 treatment reduced the expression of NF-κB downstream target genes such as *ICAM1* (16-fold), *FTH* (1.5-fold), *FAS* (2-fold), and *IRF1* (4-fold) (Figure 1G). This ability of BAY651942 to inhibit RPE–EMT increased with dose, but there were clear limitations, as at higher doses (>48 µM), BAY651942 was toxic to the RPE monolayers and caused significant RPE cell death after 24 h of treatment. As another limitation, in contrast to its potent effects in the TGF–β/TNF–α model, BAY651942 neither inhibited EMT nor restored RPE marker expression in the enzymatic dissociation induced RPE–EMT model. 

These results suggest that blockade of NF-κB signaling may be a potent pharmacological target for inhibiting RPE–EMT and maintaining RPE identity.

### 3.2. BAY651942 Restores Normalized Expression of Multiple Axon Guidance Molecules

To expand on these qPCR studies and better dissect the molecular events associated with BAY651942′s inhibition of TGF–β/TNF–α-induced RPE–EMT, we performed RNA-seq analysis on RPE monolayer cultures treated with TGF–β/TNF–α only, TGF–β/TNF–α plus vehicle (DMSO) control, and TGF–β/TNF–α plus BAY651942 (16 µM, 32 µM) for 24 h. We applied a stringent statistic threshold of greater than or equal to twofold change, FDR of <0.05, and a *p*-value of <0.05 to identify DEG between drug- and DMSO-treated samples (Figure 2A; Appendix A). As shown by principal component analysis (PCA), with the exception of one “outlier” sample (a 32 µM treated sample), there was overall consistency of the replicates and the relative position of the control, DMSO, and drug treated samples (Appendix A). Because our previous transcriptomic study [11] identified axon guidance signaling as the top enriched pathway from TGF–β/TNF–α-induced RPE–EMT, we examined the top 40 expressed axon guidance molecules from the RNA-seq dataset. Our findings reveal that multiple dysregulated axon guidance cues (*NGEF, UNC5B, NTNG1, UNC5D, SEMA3C, SEMA4B, SEMA3D, SEMA6D, EPHB1, EPHB2, EFNA5 EFNB1,* and *EFNB2*), as well as other axon guidance cues indirectly regulated through cytoskeletal dynamics (*ROBO2, LIMK1, RGS3, RHOA*), were significantly altered by at least one dose of BAY651942 treatment (Figure 2B). To confirm our RNA-seq results, we performed qPCR validation for select axon guidance genes using the same RNA samples used for the RNA-seq study and noted a tight correlation (Appendix A). These results, taken together, demonstrate that BAY651942 can reverse some of the axonal guidance-related gene expression changes associated with TGF–β/TNF–α-induced RPE-EMT (expression of *EFNB1*, *MICAL2*, *RGS3*, *BMP7*, *GNB3*, *NTNG*, *SEMA3D ITGA2,* and *ITGA5* were all significantly modulated by treatment with 32 µM BAY 651942) (Figure 2B–D).

### 3.3. BAY651942 Modulates Multiple RPE–EMT-Associated Biological Pathways and Upstream Regulators 

In addition to modulating the effects of RPE–EMT on the expression of axon guidance-related molecules, we found that BAY651942 treatment also modulated a number of other signaling pathways, including ILK signaling, BMP signaling, coagulation pathway, TGF–β signaling, osteoarthritis pathway, RAR (retinoic acid receptor) activation, and IL-8 signaling (Figure 3A, Appendix A). To investigate the cascade of transcription regulators acting upstream of the observed DEGs that occur in response to dissociation or TGF–β/TNF–α-induced EMT, we analyzed the RPE–EMT transcriptome using the “upstream regulator analysis” (URA) module in IPA. The URA algorithm uses *p*-value overlap and activation z-score to predict regulators that have already been experimentally shown to alter gene expression (“transcription regulators”) and may influence the genes identified from the input pathways. IPA enables testing of the biological processes, pathways, and diseases the transcription regulators and their targets may control. It also shows how the upstream molecules may regulate one another. These upstream molecules are predominantly DNA-associated transcription factors but include kinases, miRNAs, translational regulators, growth factors, and cytokines, as well as exogenous drugs, such as kinase inhibitors (Appendix A) [38,54]. We filtered for regulators which were predicted by IPA analysis to be activated or inhibited for at least one time point in the dissociation EMT-induced samples and one dose from TGF–β/TNF–α-induced EMT samples. We clustered the resulting URs on the basis of the absolute IPA activation score at any dissociation time/TGF–β/TNF–α dose to identify temporal and dose–dependent regulated patterns (Appendix A and Figure 3B–G). URA from BAY651942-treated RPE shows strong evidence for the inhibition of several possible EMT-associated factors including transcription factors (Figure 3B), kinases (Figure 3C), cytokines (Figure 3D), enzymes (Figure 3E), growth factors (Figure 3F), and miRNAs (Figure 3G). Together, these findings implicate several of the likely molecular mediators and mechanisms by which BAY651942 modulates TGF–β/TNF–α-induced RPE–EMT. 

### 3.4. BAY651942 Restores Multiple AMD-Associated Risk Factors That Were Altered by TGF–β/TNF–α-Induced RPE–EMT

Our earlier studies noted the expression of several genome wide association study-defined AMD-associated risk factor genes [55] were altered by treatment with TGF–β/TNF–α [11]. To extend our findings on the ability of BAY651942 to partially inhibit TGF-β/TNF–α-induced RPE–EMT, we tested whether it could modulate the effect of RPE–EMT on expression of complement genes and other AMD risk genes. We compared our RPE–EMT dataset (Figure 4A) to a previously curated list of 283 AMD-associated genes [56]. From this list, we identified 257 genes from our dissociation and TGF–β/TNF–α-induced EMT data sets. Solute carrier (SLC) groups of membrane transport proteins are crucial for visual function, as the deletion of *SLC16A8* in mice leads to vision loss [57] and decreased expression of *SLC16A8* in eyes with geographic atrophy (GA) increases disease severity [58]. Our transcriptomic analysis shows decreased expression of SLC family genes during TGF–β/TNF–α-induced RPE–EMT including zinc transporter *SLC39A12* (31-fold), a hypoxia-inducible key regulator [59], *SLC6A13* (53-fold), a sodium and chloride-dependent GABA transporter [60], and monocarboxylate transporter family, *SLC16A10* (13-fold). Interestingly, these factors were significantly restored with BAY651942 treatment (Figure 4B) indicating potential clinical significance, as *SLC16A8* and *SLC16A10* expression are altered during AMD progression [61].

Several studies have shown that elevated levels of matrix metalloproteinases (MMPs) are a hallmark of EMT progression during tumor metastasis and alter cell-cell contact leading to ECM degradation [62]. Additionally, dysregulated MMP homeostasis contributes to various retinal diseases, including PVR and AMD [63,64]. We identified several MMP family members including *MMP1* (18-fold)*, MMP3* (3.5-fold)*, MMP9* (213-fold), and *MMP10* (46-fold), which were increased during TGF–β/TNF–α-induced RPE–EMT. Of these, *MMP3* and *MMP9* have been reported as risk factors for AMD progression [65]. A previous study showed that *SNAI1* expression is regulated by the secretion of *MMP9* to induce EMT during tumor cell invasion [66]. Our RNA-seq analysis shows that the expression of MMPs, such as *MMP1*, *MMP3,* and *MMP9,* were significantly decreased with BAY651942 treatment (Appendix A), confirmed by qRT-PCR analysis (Figure 4C). We found that RPE [*BEST1*, *RPE65*, *RLBP1*, and *LRAT*] (Figure 4D) and EMT [*FN1, HMGA2, SNAI1,* and *CDH2*] (Figure 4E) factors were also partially or completely restored by BAY651942 treatment. Furthermore, AMD-associated complement factors-C5 [67], cytokines-NAMP [68] were reduced after BAY651942 treatment, supporting the efficacy of BAY651942 as a potential therapeutic molecule to reduce RPE abnormalities and dysfunction in AMD. Taken together, these findings suggest BAY651942 modulates the expression of multiple AMD-associated risk factors altered during TGF–β/TNF–α-induced RPE–EMT.

### 3.5. Validation of TGF–β/TNF–α-Induced RPE–EMT Using IKKβ Inhibitor (BMS345541)

To validate and generalize our findings with BAY651942, we tested another IKKβ inhibitor, BMS345541 [31], with RPE generated from two independent human stem cell lines, the iPS line IMR90.4 and the embryonic stem cell line H7. Consistent with BAY651942’s results, BMS345541 partially suppressed RPE–EMT and restored expression of RPE factors. Our data showed that the number of RPE factors suppressed due to TGF–β/TNF–α-induced RPE–EMT were significantly increased via BMS34554 treatment (64 µM) in hiPS (IMR90.4) derived RPE monolayers. Our qPCR analysis revealed that *BEST1* (4-fold)*, MITF* (2-fold)*, TYR* (2-fold)*, RDH5* (5-fold)*, CDH1* (2-fold) and *APOE* (4-fold) were significantly restored from BMS34554 treatment (Figure 5A–C). Next, we found elevated EMT-associated factors, such as HMGA2 (8-fold), JAG1 (4-fold), integrin members *ITGA2* (11-fold), *ITGA5* (3-fold) and matrix metalloproteinases (MMP) factors *MMP1* (3.5-fold), *MMP9* (139-fold) and NF-κB downstream target gene *ICAM1* (51-fold) were significantly decreased from BMS34554 treatment (Figure 5D–H). Together, these findings indicate that IKKβ inhibitors can partially suppress TGF–β/TNF–α-induced RPE–EMT.

### 3.6. Hierarchial Analysis of Transcription Factors Affected by Inhibition of NFκB Pathway

We performed a hierarchical analysis of the DEGs and their potential regulators to better understand their relationships. To do this, we integrated our gene expression data with chromatin accessibility in RPE and extracted the promoter sequences of DEGs located in open chromatin regions in RPE. These regions can interact with transcription factors (TFs), so we scanned them with TF binding motifs available in TRANSFAC to identify enriched TF binding motifs in up-regulated and down-regulated DEGs (Figure 6A). These motifs represent the binding sites of possible DEG regulators. This analysis identified several TFs as candidate regulators of the up-regulated DEGs, including REST, KLF10, Zfp281, TRPS1, JunD, GATA, and E2F4, of which many are known to be associated with EMT (Figure 6B). For instance, REST is a crucial regulator for acquiring EMT-like and stemness, and TRPS1 positively correlates with E-cadherin and β-catenin expression in ERα-positive breast cancer cells [69]. Additionally, we identified a set of TFs predicted to regulate the down-regulated DEGs, including NFkB, JunB, ZNF7, and ISRE [70] (Figure 6C) Furthermore, we performed a STRING analysis of the DEGs to examine their connectivity, including both direct (physical) and indirect (functional) associations. Interestingly, we found that many DEGs form a network, suggesting that they cooperate during EMT. Notably, MYC and Sox17 were identified as hub genes in the network with high connectivity to other DEGs (Figure 6D). In summary, our hierarchical analysis identifies potential regulators of DEGs and their relationships, highlighting the importance of cooperation among DEGs as EMT takes place.

## 4. Discussion

Our prior transcriptome study identified multiple upstream regulators that can modulate TGF–β/TNF–α-induced RPE–EMT, and among them were several factors in the NF-κB signaling pathway, including IKKβ [11,71]. Recently, we have also found that the IKKβ inhibitor BAY651942 can protect RPE from sodium iodate-induced oxidative stress in mice [72]. In this current study, we identified BAY651942 as a small molecule suppressor of TGF–β/TNF–α-induced RPE-EMT. We dissected the molecular events and associated biological pathways altered by BAY651942 treatment during RPE–EMT progression using unbiased genome-wide transcriptome analysis to broadly characterize IKKβ’s role in modulating RPE–EMT. We validated the effects of BAY651942 using another IKKβ inhibitor, BMS345541, in independent hRPE lines. Our results, taken together, indicate IKKβ inhibition attenuates gene expression changes associated with TGF–β/TNF–α-induced RPE–EMT, including alterations in the expression of multiple AMD-associated risk factors. Below, we discuss some of the implications and translational potential of these findings.

Stress and injury induce various pathological changes in the RPE, including de-differentiation and EMT, which lead to RPE dysfunction and potentially cell death [4,5,73]. RPE–EMT contributes to the formation of fibrous epiretinal membranes (ERMs) in PVR [6,7,13], and the contractile activity of affected RPE cells can cause complex retinal detachments [6]. Additionally, oxidative stress, which is one of the major risk factors for developing AMD [74], can cause EMT–like changes of RPE cells [75,76,77]. Of additional interest, increasing data suggests that RPE–EMT may contribute to dry and wet AMD [8,14,15,16]. Increased SNAI1 and vimentin with decreased E-cadherin were observed in the RPE/choroid of human dry AMD eyes compared to age-matched controls [8]. Of further interest and potential clinical relevance, hypertrophic RPE was noted in the atrophic region of human eyes with geographic atrophy (GA), an advanced form of dry AMD, as well as around the edge of choroidal neovascularization (CNV) in human eyes with wet AMD [78]. However, whether RPE hypertrophy in these GA and CNV eyes is related to EMT is unclear and requires further investigation.

To study RPE–EMT, the co-treatment of hRPE cultures with TGF–β/TNF–α, which synergistically activates an EMT program compared with treatment with either cytokine alone [7,10,11,12], has been used. In our investigation, this model proved useful in conducting molecular-level dissection of the RPE–EMT pathways. Notably, BAY651942 treatment attenuated expression changes of multiple EMT-related genes as well as genes important to RPE integrity and mature functions, such as those involved in the visual cycle and melanin pigment synthesis. Most importantly, BAY651942 treatment also partially restored the expression of multiple AMD-associated risk factors that were altered during TGF–β/TNF–α-induced RPE–EMT, including complement factors, matrix metalloproteinases, solute carrier transport proteins, and pro-inflammatory cytokines. These results show that IKKβ inhibition not only attenuates EMT-related gene expression patterns, but also preserves the normal expression levels of AMD-associated risk factors in hRPE cells. This suggests that the inhibition of IKKβ could potentially be used as a therapeutic approach for retinal diseases that involve RPE–EMT, such as PVR and AMD. 

As described above, our previous studies predicted the NF-κB signaling pathway as one of the potential upstream regulators of RPE–EMT induced by TGF–β/TNF–α in hiPS-RPE cells [11]. To activate NF-κB, which is normally associated with IκB in the cytoplasm in the resting state, IκB needs to be phosphorylated and degraded, which enables NF-κB dimers to translocate to the nucleus and function as a DNA-binding transcription factor [20]. The key regulatory node of the NF-κB cascade is the IκB kinase (IKK) complex on which diverse stimuli and inputs from other signaling pathway converge [21,22,23,24]. To test the role of NF-κB, past studies have targeted one of the IKK catalytic subunits, IKKβ, by pharmacological inhibition and genetic ablation [29,30,79]. In these reported studies, the selective IKKβ inhibitor, BAY651942, suppressed mostly inflammation-related pathological phenomena, such as pulmonary inflammation in animal models of asthma, edema formation in the ear caused by chemical inducers [29], myocardial injury following acute ischemia-reperfusion injury [80], and lipopolysaccharide-induced neurotoxicity to dopamine neurons in a Parkinson’s disease model [30]. Since NF-κB is a master regulator of inflammation and immune responses, these reported effects of BAY651942 are reasonable and expected. In our study, using pathway analysis, we found multiple lymphokine signaling pathways were altered by BAY651942 treatment. In addition, a number of individual cytokines were implicated by upstream regulator analysis. These findings suggest that inflammatory components are likely involved in our RPE–EMT system. However, we also found alterations or restorations of the expression of multiple EMT–related genes by BAY651942 treatment. Thus, we speculate that the effects of BAY651942 are of a wider biological spectrum, involving more than just inflammation and modulation of immune reactivity. It is also interesting to note that, while mice with conditional knockout of *Ikbkb* (also known as *Ikk2*) in their retina showed no obvious defects in retinal development or function, they did show significantly reduced laser-induced CNV [24].

EMT can be activated by multiple signaling pathways, such as TGF–β, Wnt/β-catenin, and Notch pathways, which are all interconnected. The TGF-β pathway is linked to the PI3K-AKT pathway, which in turn triggers the activation of several signaling pathways, including the NF-κB pathway [17,81]. These pathways ultimately lead to the transcriptional activation of EMT–TFs, such as SNAI1, ZEB1, and TWIST, in the nucleus. These EMT-TFs regulate their target genes to activate EMT programs at the cellular level by upregulating mesenchymal cell-related factors, such as FN1 (fibronectin), CDH2 (N-cadherin), and MMPs, and, in parallel, downregulate epithelial cell markers, such as CDH1 (E-cadherin) [18]. EMT is a complex cellular process with many different variations and properties, which prompted a ‘Consensus Statement’ from the EMT International Association (TEMTIA) [82]. Likely reflecting such diversity, we observed that blocking TGF–β/TNF–α-induced RPE–EMT by IKKβ inhibitors resulted in the restoration of gene expression patterns; however, we found variability in the degree of restoration, i.e., the expression levels were nearly completely restored for some genes but only partially for others. In addition, each gene responded differently to different concentrations of compounds. The complex patterns of IKKβ inhibitors’ effects also likely reflect the complexity of the EMT regulatory pathways and their interconnected networks. Further reflecting the complexity of RPE biology, RPE cells have capacities to dedifferentiate or transdifferentiate under certain conditions or under stress [77,83,84,85]. The unbiased transcriptomic data presented here will, we hope, help advance the characterization and understanding of RPE–EMT biology and disease mechanisms.

In summary, our results show that IKKβ inhibitors restore gene expression patterns altered during TGF–β/TNF–α-induced RPE–EMT in hiPS-RPE cells, including those with multiple AMD-associated risk factors. Thus, our results identify the inhibition of IKKβ as a potential approach to treating retinal diseases that involve RPE–EMT, possibly including AMD.

## Figures and Tables

**Figure 1 cells-12-01155-f001:**
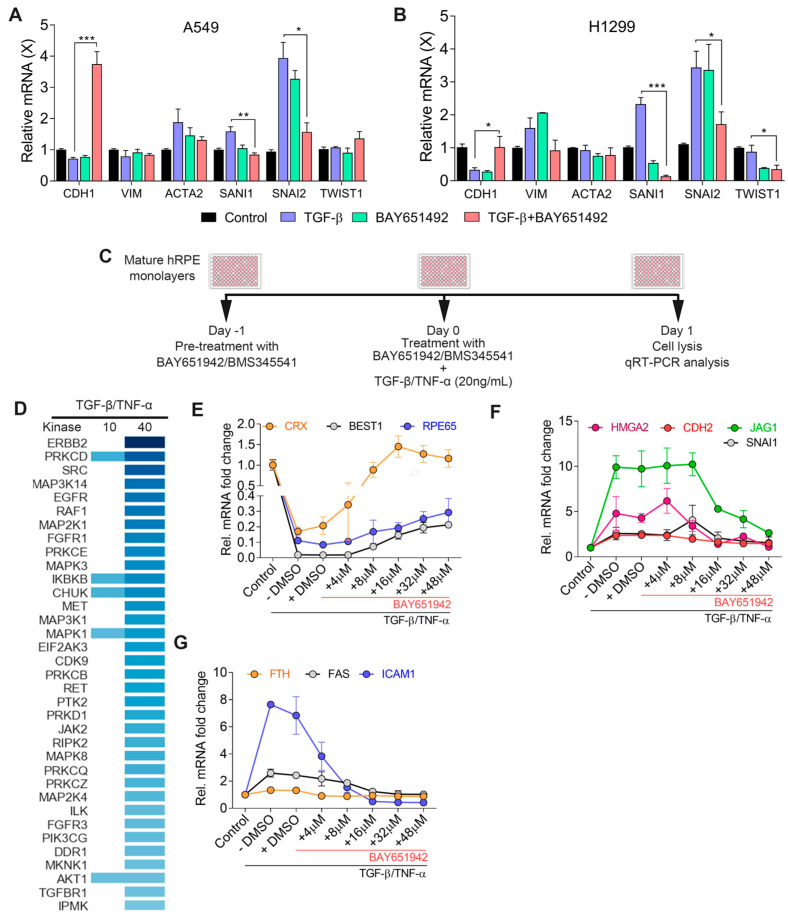
Identification of small molecule kinase inhibitors that modulate TGF–β/TNF–α associated RPE–EMT. qPCR validation of altered EMT-associated factors in (**A**) non-small cell lung cancer A549 and (**B**) H1299 cells. (**C**) Schematic representation of kinase inhibitors treatment and RPE–EMT induction. (**D**) Heatmap of potential small molecule kinase inhibitors identified by IPA-generated upstream regulator analysis. qPCR analysis shows differential expression of (**E**) RPE genes, (**F**) EMT and (**G**) NF-κB factors from TGF–β/TNF–α-induced EMT were altered by BAY651942 treatment. Error bars represent SD of at least three biological replicates and statistically significant mean differences (*p* ≤ 0.05 by ANOVA). * *p* ≤ 0.05; ** *p* ≤ 0.01; *** *p* ≤ 0.001.

**Figure 2 cells-12-01155-f002:**
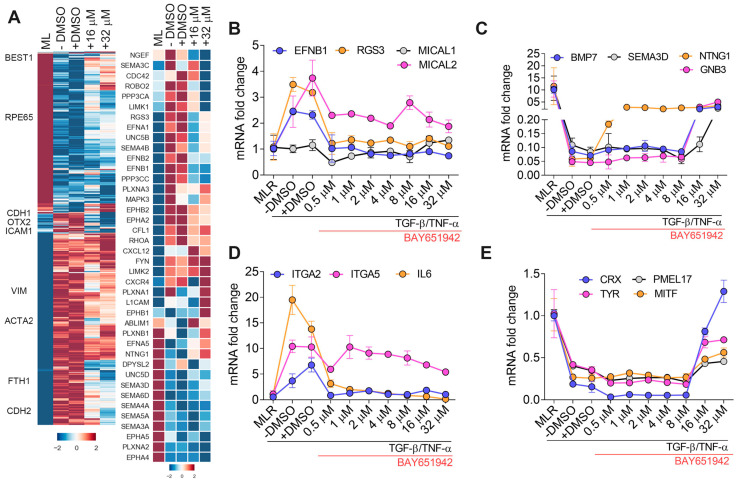
Transcriptomic analysis of BAY651942 modulated RPE–EMT in TGF–β/TNF–α model. (**A**) Hierarchical clustered heatmaps of log2-transformed ratios. Average abundances of DEG showing significant differences from DMSO and BAY651942 treatment after TGF–β/TNF–α-induced EMT in hiPS–RPE. Heatmap of axon guidance genes that were altered by BAY651942 treatment compared TGF–β/TNF–α plus DMSO treatment. (**B**–**E**) qPCR validation of altered multiple axon guidance genes (**B**–**D**), RPE genes (**E**) by BAY651942 treatment followed by TGF–β/TNF–α-induced EMT. Error bars represent SD of at least three biological replicates and statistically significant mean differences (*p* < 0.05 by ANOVA).

**Figure 3 cells-12-01155-f003:**
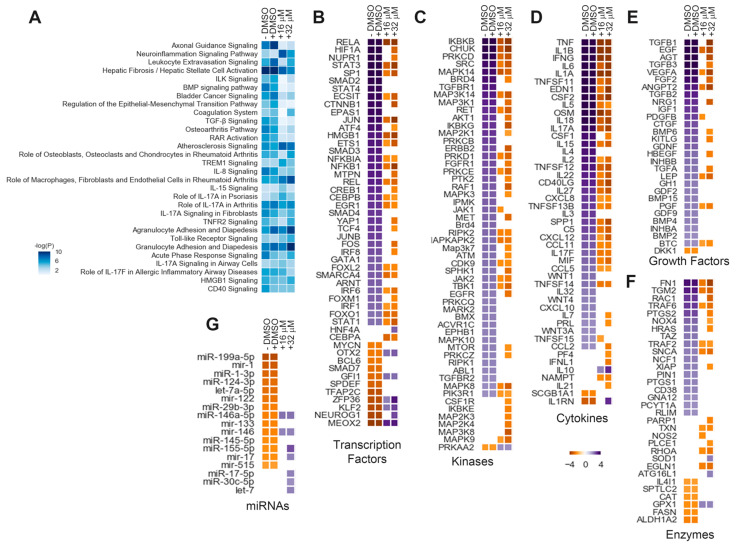
BAY651942 regulated transcriptome dynamics of TGF–β/TNF–α-induced RPE. (**A**) Top canonical pathways were predicted based on the alteration of highly-enriched genes that changed in abundance (activated or inhibited) from DMSO and BAY651942 treated hiPS-RPE monolayers prior to TGF–β/TNF–α-induced EMT (**B**–**G**) heatmaps of IPA-generated upstream regulator analysis from DMSO- and BAY651942-treated hiPS RPE monolayers. IPA uses activation Z-score as a statistical measure of the match between expected relationship direction and observed changes in the gene expression regulator such as different transcription factors; (TFs) (**B**), kinases (**C**), cytokines (**D**), growth factors (**E**), enzymes (**F**), and miRNAs (**G**) were predicted to be activated (violet) or inhibited (yellow) after DMSO and BAY651942. Z-scores of >2 (activated) or <−2 (inhibited) were considered significant. Only genes with statistically significant changes at an FDR of 5% (*p* < 0.05) were included in the analysis.

**Figure 4 cells-12-01155-f004:**
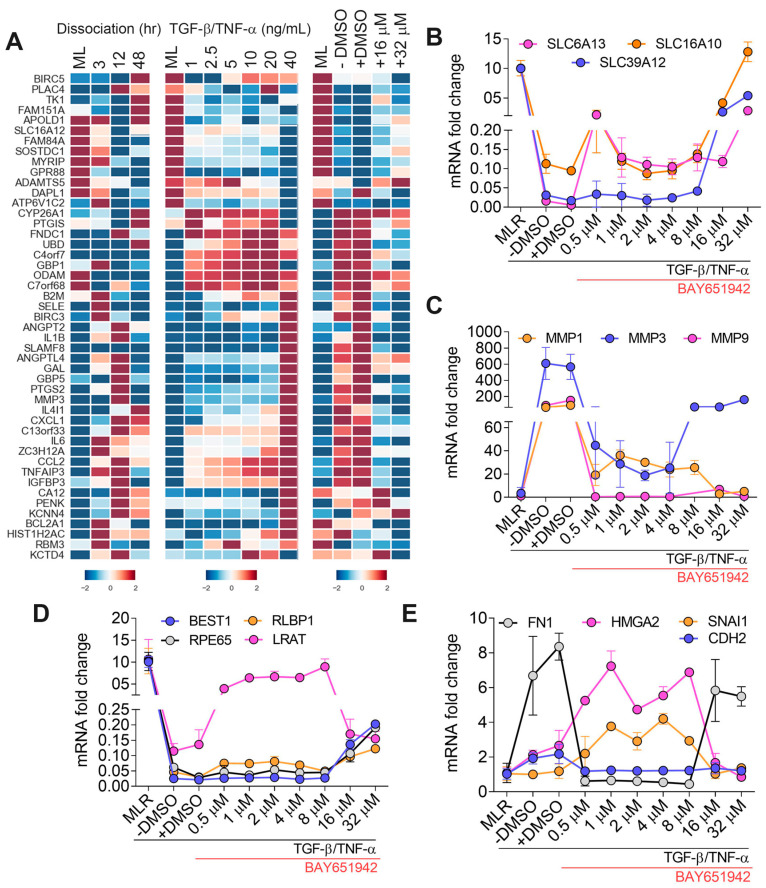
Differential expression of AMD-associated risk factor modulated by BAY651942 treatment. (**A**) Heatmaps of dysregulated AMD-associated risk factors altered due to dissociation, TGF–β/TNF–α-induced EMT and differentially regulated with the treatment for BAY651942 from TGF–β/TNF–α. (**B**–**E**) qPCR validation of multiple AMD-associated risk factors (**B**,**C**), RPE factors and EMT factors (**D**) by BAY651942 treatment followed by TGF–β/TNF–α-induced EMT. Error bars represent SD of at least three biological replicates and statistically significant mean differences (*p* < 0.05 by ANOVA).

**Figure 5 cells-12-01155-f005:**
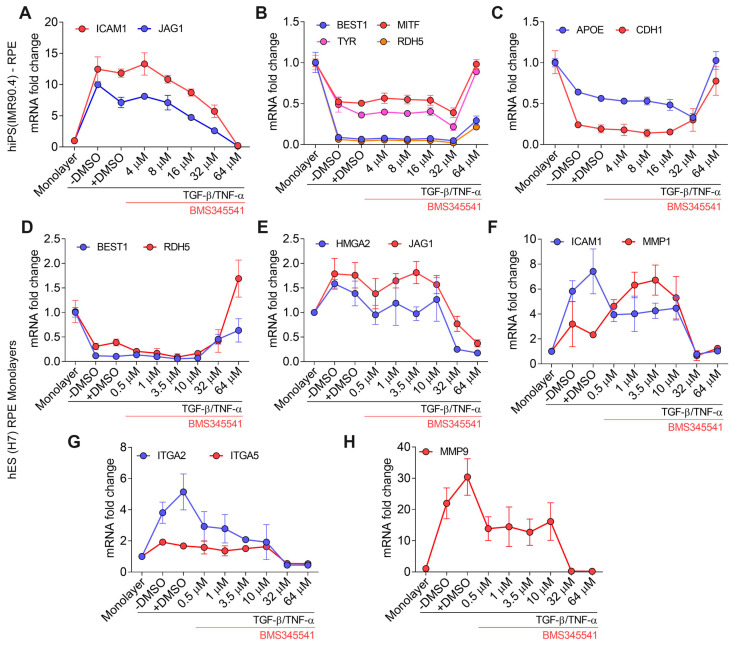
qPCR validation of TGF–β/TNF–α-induced RPE–EMT by IKKβ inhibitor BMS345541. (**A**–**C**) Differential expression of RPE-EMT-associated factors were measured after treatment with BMS345541 in TGF–β/TNF–α-induced EMT in hiPS (IMR90.4) RPE. (**D**–**H**). Differential expression of RPE–EMT-associated factors were measured after treatment with BMS345541 in TGF–β/TNF–α-induced EMT in hES (H7) RPE.

**Figure 6 cells-12-01155-f006:**
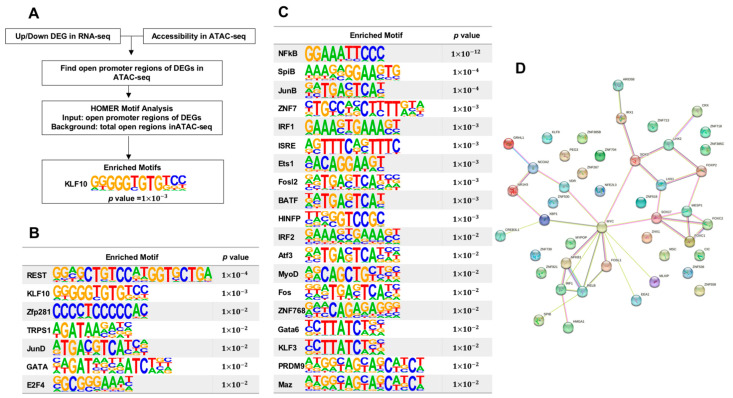
Hierarchical analysis of transcription factors impacted by NFκB pathway inhibition. (**A**) Flowchart for identifying potential regulators of differentially expressed genes (DEGs). (**B**) Enriched binding motifs from up-regulated DEGs. (**C**) Enriched binding motifs from down-regulated DEGs. (**D**) Connectivity diagram of DEGs.

## Data Availability

All raw sequencing files and expression matrices generated under this study can be found at GEO under the accession number GSE228934.

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
