# Peer review of "IKKβ Inhibition Attenuates Epithelial Mesenchymal Transition of Human Stem Cell-Derived Retinal Pigment Epithelium"

_cells, 2023, doi:10.3390/cells12081155_

Round 1
Reviewer 1 Report
This study evaluates NFkB pathway inhibitors for prevention of epithelial to mesenchymal transition (EMT) in retinal pigment epithelial cells (RPE). This is important because RPE EMT occurs is human diseases including age-related macular degeneration and proliferative vitreoretinopathy. Prior studies by these authors showed that iPS-RPE cells undergo EMT when stimulated with TGF-beta/TNF-alpha and transcriptomic analysis implicated the NFkB pathway. Herein, elegant experiments using two different pharmacological inhibitors of NFkB in RPE derived from two different stem cell lines partially inhibited the gene expression signature of EMT and maintained the differentiated RPE transcriptome. The only suggestion for improvement is to perform a hierarchical analysis of transcription factors affected by NFkB inhibition in order to better understand which are upstream drivers of the EMT phenotype in this model.
Author Response
Response to reviewers’ comments, February 22, 2023.
Revision of Manuscript: Cells-2121403
Title: "IKKβ inhibition attenuates epithelial mesenchymal transition of human stem cell derived retinal pigment epithelium"
Dear editor and reviewer,
We would like to thank you all for your reviews and your constructive comments related to our submitted manuscript (Cells-2121403). Based upon your comments and suggestions, we have performed additional analysis and also revised the text. We feel that the additional work we have been able to accomplish has addressed most of the points raised by the reviewers, has significantly improved our manuscript, and we hope that you will agree. Please find below our point-by-point description of how we have tried to respond to each of the reviewers’ individual comments and concerns.
Reviewer #1 Comments:
Reviewer 1: This study evaluates NFkB pathway inhibitors for prevention of epithelial to mesenchymal transition (EMT) in retinal pigment epithelial cells (RPE). This is important because RPE EMT occurs is human diseases including age-related macular degeneration and proliferative vitreoretinopathy. Prior studies by these authors showed that iPS-RPE cells undergo EMT when stimulated with TGF-beta/TNF-alpha and transcriptomic analysis implicated the NFkB pathway. Herein, elegant experiments using two different pharmacological inhibitors of NFkB in RPE derived from two different stem cell lines partially inhibited the gene expression signature of EMT and maintained the differentiated RPE transcriptome.
<Response> Thank you for your positive feedback on our manuscript. We are pleased to hear that our study provides valuable resources for investigators studying the RPE-EMT phenomenon, including those studying age-related macular degeneration and proliferative vitreoretinopathy disease models. We hope that our work will contribute to a better understanding of the complexities of EMT and its regulation and its role in these and other diseases.
The only suggestion for improvement is to perform a “hierarchical analysis of transcription factors affected by NFkB inhibition in order to better understand which are upstream drivers of the EMT phenotype in this model”
<Response> We thank the reviewer for this suggestion, which has helped to provide additional insights for our study. We have performed a hierarchical analysis of transcription factors affected by NF-kB/IKKB inhibition, and we have included a new figure (Fig. 6) in the revised manuscript to illustrate our findings. Additionally, we have added a new methods section (Methods Section 2.9) and results section (Results Section 3.6) in the revised manuscript to provide a more detailed description of our approach and results. We appreciate your valuable feedback and hope that these changes will further enhance the quality and clarity of our work.
Reviewer 2 Report
This is a very well thought out, executed and written study by Sripathi et al describing small molecule inhibition of NF-kB pathway helping to alter EMT of cultured RPE monolayers. Below are my comments/suggestions.
Major Comments
1. In Figure 1H the intensity of RPE65 immunostaining is much brighter in BAY651942-treated RPE than control monolayer RPE. Were these conditions all imaged at the same time using the same paramaters (i.e., same fluorescent laser intensity) or does BAY651942 treatment actually increase expression of RPE65? Do monolayers treated with BAY651942 alone also exhibit increased RPE65 expression compared to untreated control monolayers? Can the authors also include a secondary antibody only control panel?
2. My biggest critique of this manuscript is that besides Figure 1H (commented on above), the data in the manuscript are all at the transcriptional level. While the data are solid and very interesting, for me the paper would be that much more meaningful with the addition of data at the protein level for a handful of important EMT markers. Whether the data is presented in the form of Western blotting or more immunocytochemistry I will leave up to the authors, but I think something along these lines is necessary to elevate the paper. I realize many of these experiments were performed using 384-well plates but perhaps the authors could choose the optimal timepoint and dose and show protein data?
Minor Comments
1. Please peruse the document for use of either “RPE EMT” or “RPE-EMT.” Please choose your preferred one and make consistent throughout.
2. Since previously established/reported, please add references of RPE EMT induction by TGFB1 and TNFa to the methods section entitled “RPE-EMT induction in vitro.”
3. Can authors please elaborate on what exactly is meant by “the geometric mean of three housekeeping genes” in the methods section for qRT-PCR?
4. On Page 8 near the end of the first paragraph there is a repeated typo error that says “URA from BAY651942 treated RPE shows the shows strong evidence…” Please delete “the shows” so the sentence makes sense.
Author Response
Response to reviewers’ comments, February 22, 2023.
Revision of Manuscript: Cells-2121403
Title: "IKKβ inhibition attenuates epithelial mesenchymal transition of human stem cell derived retinal pigment epithelium"
Dear editor and reviewers,
We would like to thank you all for your reviews and your constructive comments related to our submitted manuscript (Cells-2121403). Based upon your comments and suggestions, we have performed additional analysis and also revised the text. We feel that the additional work we have been able to accomplish has addressed most of the points raised by the reviewers, has significantly improved our manuscript, and we hope that you will agree. Please find below our point-by-point description of how we have tried to respond to each of the reviewers’ individual comments and concerns.
Reviewer #2 Comments:
Major Comments
- In Figure 1H the intensity of RPE65 immunostaining is much brighter in BAY651942-treated RPE than control monolayer RPE. Were these conditions all imaged at the same time using the same paramaters (i.e., same fluorescent laser intensity) or does BAY651942 treatment actually increase expression of RPE65? Do monolayers treated with BAY651942 alone also exhibit increased RPE65 expression compared to untreated control monolayers? Can the authors also include a secondary antibody only control panel?
<Response> We thank the reviewer for raising this important and critical point. We agree that the immunostaining is much brighter in BAY651942-treated RPE than control monolayer RPE, presented in our original Figure 1H. In fact, frankly, we are a bit embarrassed that we did not ourselves pick up this discrepancy. Yes, we used the same fluorescent laser intensity/exposure parameters for the imaging. However, without repeating these experiments several times using more than one independent hiPS-RPE line, with other RPE specific factors, we cannot be confident of the generality of our result. We thus understand that repeated immunostaining data would be helpful in further validating our findings. However, generating this new data, including staining for mature RPE marker RPE65 in hiPS-RPE in multiple lines would require many months, as a new batch of hiPS-RPE differentiation from stem cells, which\takes several months, would have to be initiated. Moreover, hiPS-RPE cells requires longer culture conditions to observe RPE65 expression. An additional challenge in performing these experiments is that the first author, Srinivasa R. Sripathi, who performed all the RPE culture work, is in the process of leaving Johns Hopkins to establish his own lab as a new faculty member at The Retina Foundation (Dallas, TX). Although, we understand this is an unusual and sub-optimal response, given these time considerations and the uncertainty about the RPE65 staining results, we have deleted the figure in the revised manuscript. Nonetheless, we will continue to explore opportunities to generate immunocytochemical analysis on BAY651942 treated RPE cells that will further support our findings in the future.
- My biggest critique of this manuscript is that besides Figure 1H (commented on above), the data in the manuscript are all at the transcriptional level. While the data are solid and very interesting, for me the paper would be that much more meaningful with the addition of data at the protein level for a handful of important EMT markers. Whether the data is presented in the form of Western blotting or more immunocytochemistry I will leave up to the authors, but I think something along these lines is necessary to elevate the paper. I realize many of these experiments were performed using 384-well plates but perhaps the authors could choose the optimal timepoint and dose and show protein data?
<Response> We appreciate the reviewers' suggestions regarding the extension of our studies at the protein level, and agree that additional protein expression data would be helpful in further validating and expanding our findings. However, due to the time considerations noted above together with the unfortunate timing of Dr. Sripathi now leaving to take up his new position at The Retina Foundation, we are not able to perform these requested experiments in a timely manner. We regret this less than satisfactory response, but hope that you will agree with us that the RNA expression data and its analysis as presented in the revised manuscript, even without additional protein data, is still of high value to the research community and worthy of publication in its present form.
Minor Comments
- Please peruse the document for use of either “RPE EMT” or “RPE-EMT.” Please choose your preferred one and make consistent throughout.
<Response> We appreciate the reviewer's suggestion, and we agree that it is important to maintain consistency in the use of terminology in our manuscript. Therefore, we have carefully revised the text to ensure that we use "RPE-EMT" consistently throughout the manuscript.
- Since previously established/reported, please add references of RPE EMT induction by TGFB1 and TNFa to the methods section entitled “RPE-EMT induction in vitro.”
<Response> As per reviewer’s suggestion, we have included the reference of our previously published work (Sripathi et al, IOVS 2021) in the revised manuscript.
- Can authors please elaborate on what exactly is meant by “the geometric mean of three housekeeping genes” in the methods section for qRT-PCR?
<Response> In the revised manuscript, we have provided a detailed explanation of how we calculated the geometric mean of housekeeping genes in the methods section and included a reference. This clarification will help readers better understand our methodology and the rationale behind our approach.
- On Page 8 near the end of the first paragraph there is a repeated typo error that says “URA from BAY651942 treated RPE shows the shows strong evidence…” Please delete “the shows” so the sentence makes sense.
<Response> Thank you for bringing this error to our attention. We have made the necessary corrections to the typo in the revised manuscript
Round 2
Reviewer 2 Report
Although I stand by my original critique that the paper would be much stronger and biologically relevant if protein data were analyzed, particularly for EMT markers, I also empathize with the lead and senior author that they are trying to wrap up this manuscript/project in a timely fashion as the lead author is moving on from the current lab. There is a substantial amount of work presented herein and I am happy to suggest publication. Congrats to the authors and best of luck to Dr. Sripathi on future endeavors running their own lab.
Author Response
Although I stand by my original critique that the paper would be much stronger and biologically relevant if protein data were analyzed, particularly for EMT markers, I also empathize with the lead and senior author that they are trying to wrap up this manuscript/project in a timely fashion as the lead author is moving on from the current lab. There is a substantial amount of work presented herein and I am happy to suggest publication. Congrats to the authors and best of luck to Dr. Sripathi on future endeavors running their own lab
<Response>I am glad to see the reviewer's positive response regarding the amount of work presented in our manuscript. I would like to express my sincere gratitude for their suggestion to publish our manuscript in Cells. Thank you for taking the time to review our work and for providing valuable feedback.